# Evaluation of risk promoting effects for age-related macular degeneration by estradiol

**Inga-Marie Pompös**[1], **Dietrich Polenz**[2], **Norbert Kociok**[1], **Silvia Winkler**[1], **Olaf Strauß**[1]*

**1** Experimental Ophthalmology, Department of Ophthalmology, Charité - Universitätsmedizin Berlin, Corporate Member of Freie Universität, Berlin Institute of Health, Humboldt-Universität zu Berlin, Berlin, Germany, **2** Department of Surgery, Experimental Surgery, Charité - Universitätsmedizin Berlin, Berlin Institute of Health and Humboldt-Universität zu Berlin, Berlin, Germany

* olaf.strauss@charite.de

## Abstract

Early menopause increases the risk for age-related macular degeneration (AMD), the most common cause of vision loss in industrialized countries. The supplementation with estradiol reduces the risk in these cases and suggesting that estradiol deficiency is a mediator of the risk association. We investigated rat models of estradiol deficiency mimicking either biological ageing (22 months of age) or early menopause by ovariectomy and age of 22 months. Serum analysis of gonadal hormones in both models showed the expected reduction in estradiol levels compared to 6 months old controls but also increases in progesterone, corticosterone and dehydroepiandrosterone sulfate (DHEA-S). Comparing the two estradiol deficiency models, we found no differences except for DHEA-S that were reduced in ovariectomized rats. The hormone status was associated with degenerative changes in the retina with higher activity of mononuclear phagocytes and p16/p21-dependent senescence. Mainly the estrogen receptor beta (ERβ) expressing cells were affected by estradiol deficiency: ganglion cells, cells of the inner nuclear layer (INL) and retinal pigment epithelial cells. An exception are photoreceptors that were ERβ negative, showed stronger degeneration in ovariectomized rats compared to sham treated animals. We conclude that either biological or ovariectomy induced estradiol deficiency might not cause but rather promote mechanisms that lead to AMD. The phenotype depends on a broader spectrum of altered hormones than on estradiol alone. Photoreceptor degeneration and cellular senescence that were ERβ independent in ovariectomized rats suggest non-estradiol effects to increase AMD risk by early menopause.

## Introduction

Age-related macular degeneration (AMD) is the third leading cause of blindness worldwide [1] and it affects approximately 195.6 million people [2]. The prevalence of

---

**Data availability statement:** All raw data files are available from the zenodo database (accession number https://doi.org/10.5281/zenodo.17464743).

**Funding:** The work was supported by governmental funding agency "Deutsche Forschungsgemeinschaft" DFG under the registration number of RE 3924/2-1 (OS; https://www.dfg.de/) and eye research specialized foundation Dr. Werner Jackstädt-Stiftung with no registration number (IMP; https://www.jack-staedt-stiftung.de/) and Charités' inhouse grant program to foster high-quality animal research strategies under the name "Charité 3R Replace-Reduce-Refine" (DP; https://charite3r.charite.de/) with no specific registration number. The funders had no role in study design, data collection and analysis, decision to publish, or preparation of the manuscript.

**Competing interests:** The authors have declared that no competing interests exist.

AMD is significantly higher in high-income regions, such as Western Europe, where it accounts for more than half of all blindness cases [3]. Due to demographic shifts, the incidence of AMD is projected to rise significantly. The WHO estimates that the number of affected individuals will exceed 243 million within the next decade, posing a significant socioeconomic burden [2].

AMD is characterized by chronic para-inflammation, a persistent low-grade inflammatory state that does not primarily serve to combat pathogens. This para-inflammation develops when the retinal pigment epithelium (RPE) loses its ability to maintain the immune privilege of the retina. This loss of immune barrier properties by the RPE results from phototoxic stress, lipofuscin accumulation, drusen formation, and decreased Bruch's membrane permeability, which progressively damage the RPE [4,5]. Vision loss occurs in the advanced stages of AMD, either through choroidal neovascularization (CNV or neovascular AMD, nAMD) or geographic atrophy (GA). Currently, no effective treatment exists for GA. For CNV, intravitreal injections of inhibitors for vascular endothelial growth factor-A (VEGF-A) have become standard clinical practice, significantly improving vision. However, recurrences and cases of non-responsiveness are frequently observed in clinical practice [6]. Overall, the anticipated increase in AMD cases highlights a growing gap in therapeutic options.

In addition to age, major risk factors for AMD include genetic predisposition, hypertension, smoking, and an increased body mass index (BMI) [7,8]. Recent studies indicate that older women are also at a higher risk of developing AMD, as supported by findings from the three major epidemiological AMD studies: the Beaver Dam Eye Study, the Blue Mountain Eye Study, and the Rotterdam Study of The Elderly [9]. Interestingly, earlier onset of sexual maturity and later onset of menopause are associated with a reduced risk of developing AMD in women [10,11]. The resulting assumption that estrogen levels play a crucial protective role was confirmed in 1992 by the Nurses' Health Study. This large case-control study found that postmenopausal estrogen supplementation reduced the risk of nAMD by 48% [12,13]. Subsequent studies have supported these findings, showing that postmenopausal hormone therapy decreases the risk of both nAMD and GA [14–17].

Estrogen is a crucial sex hormone in both women and men, with significant extragonadal synthesis. In postmenopausal women, estradiol—the most biologically active estrogen—can decrease by up to 80%, leading to heightened inflammatory activity and oxidative stress. Postmenopausal estrogen deficiency correlates with elevated levels of pro-inflammatory cytokines, including interleukin-1β (IL-1β), monocyte chemotactic protein-1 (MCP-1), interleukin-8 (IL-8), interleukin-6 (IL-6), and tumor necrosis factor-α (TNFα), as well as increased production of reactive oxygen species (ROS) and lipid peroxidation end-products such as malondialdehyde (MDA), hydroxynonenal (HNE), and advanced glycation end-products (AGE) [17,18]. Hormone therapy during menopause or after ovariectomy reduces the levels of pro-inflammatory cytokines [19–21] and protects against excessive ROS release [22,23].

Being the blood/retina barrier, the RPE is under influence of systemically circulating steroid hormones such as mineralocorticoids or aldosterone [24,25]. The effects

of reduced systemic estrogen levels on the RPE remain incompletely understood. However, estrogen appears to be a protective factor against AMD by mitigating oxidative stress and para-inflammation. The primary nuclear and membrane estrogen receptors (ERα, ERß, PGR30) are also expressed in the retina, particularly in the RPE. This presence is consistent across ages and does not differ between genders [14,15].

Under physiological conditions, the RPE ensures the maintenance of the immune-privilege of the retina. It is a crucial part of the outer blood-retinal barrier, with the fenestrated choroidal endothelium and the RPE as tight epithelium creating the border through impermeable tight junctions. At this regulatory checkpoint, the RPE inhibits local immune responses [5,26]. The RPE is capable of secreting immunosuppressive or immunoregulatory factors and expressing surface molecules with immunoregulatory or cytotoxic functions; in addition, it has anaphylatoxin receptors, toll-like receptors, IL-1ß and TNFα receptors, among others, for efficient exchange with immune cells. Effector cells (from the innate and adaptive immune system) are converted by the RPE into a regulatory or reparative phenotype when this barrier is crossed [27]. In a disease state, this immunogenic function of the RPE is disturbed. The disturbance of immunogenic function precedes decades of long harmful metabolic/photo-oxidative impacts on the RPE [4,28–32]. The subsequent loss of RPE cells does not lead to impairment of the monolayer but evokes local inflammation through complement activation and attraction of mononuclear phagocytes, Iba1+ cells [33–35]. Complement and immune cells overcome the RPEs' immune barrier properties, the latter ones by TNFα and IL-1β secretion [33,34]. The phenotype of the RPE changes into one with immunostimulatory and pro-inflammatory properties. By secreting MCP-1 and complement factors, and other cytokines and chemokines such as IL-6 and VEGF-A, the RPE attracts and activates facilitating their transition into a pro-inflammatory phenotype that infiltrates the retina [5,36,37]. The precise cause of this switch from the immunomodulatory to the pro-inflammatory phenotype of RPE is still unclear. We hypothesize that systemic estrogen deficiency compromises the immune-inhibitory phenotype of the RPE, thereby influencing the risk of nAMD.

To test this hypothesis, we investigated two rat models for estradiol deficiency: 22 months old rats with biological estradiol deficit and rats ovariectomized at the age of 8 months and investigated at 22 months of age as well; 6 months old rats served as controls. The rat retinas were examined using quantitative PCR (qPCR), immunocytochemistry, and AI-based image analysis. The results were assessed on account of the serum levels of gonadal hormones. In addition, functional tests of ERβ expression were conducted *in vitro* using ARPE-19 cells. These experiments demonstrated structural degeneration in estradiol receptor-positive retinal cells, increased Iba1+ cell activity, and abundant senescence markers in estrogen-deficient rat models. Serum analyses indicated that these changes were not exclusively attributable to estradiol deficiency, suggesting that additional gonadal hormones may contribute to disease pathology.

## Materials and methods

### *In vitro* experiments; ARPE-19 cell culture and PCR analyses

ARPE-19 cells were cultured in Dulbecco's Modified Eagle Medium/F12 with a supplement of 10% Fetal Bovine Serum and 1% Penicillin/Streptomycin. The culture was maintained at 37°C with a controlled level of 5% $CO_2$. Prior to the experiment, semi-confluent (80% confluent) cells were incubated in serum-free medium overnight. The cells were exposed to 17ß-Estradiol 4µM for 6 hours and TNFα 0.6nM for 2 hours (Table 1). The isolation of RNA was carried out using the RNeasy Plus Mini Kit, followed by the synthesis of complementary DNA (cDNA) using the QuantiTect Reverse Transcription Kit. Primers for quantitative polymerase chain reaction (qPCR) were obtained from Eurofins Genomics. The primer sequences are provided in Table 2. Gene expression levels were quantified by qPCR using a Biozym SYBR green PCR kit on a RotorGene as a real-time PCR cycler. The ΔΔCT method was used for analysis. The expression of ERα (*ESR1*) and ERß (*ESR2*) was confirmed using a RT-PCR, followed by the separation of the samples using 2% agarose gel electrophoresis with ethidium bromide.

**Table 1. Material information for *in vitro* experiments.**

| Material | Source | Cat.No |
|---|---|---|
| ARPE-19 cells | ATCC, Manassas, USA | CRL-2302 |
| 17ß-Estradiol | Sigma-Aldrich, Darmstadt, Germany | 50-28-2 |
| TNFa | R&D Systems, Minneapolis, USA | 210-TA-020/CF |
| RNeasy Plus Mini Kit | Qiagen, Hilden, Germany | 74136 |
| QuantiTect Reverse Transcription Kit | Qiagen, Hilden, Germany | 205314 |
| SYBR green PCR kit | Biozym, Oldendorf, Germany | 331416S |

**Table 2. primers used for *in vitro* PCR analyses.**

| Primer | Forward | Reverse | Product Length | Template |
|---|---|---|---|---|
| GAPDH | TCAACGACCACTTTGTCAAGCTCA | GCTGGTGGTCCAGGGGTCTTACT | 119 bp | NM_001357943.2 |
| IL8 | ACT CCA AAC CTT TCC ACC CC | TTC TCA GCC CTC TTC AAA AAC T | 175 bp | NM_000584.4 |
| CFH | CCT GAT CGC AAG AAA GAC CAG | ACT GAA CGG AAT TAG GTC CAA C | 94 bp | NM_000186.4 |
| VEGFA | CCAGCACATAGGAGAGATG | GGAACATTTACACGTCTGC | 223 bp | NM_003376.6 |
| IL1b | TCG CCA GTG AAA TGA TGG CT | TGG AAG GAG CAC TTC ATC TGT T | 91 bp | NM_000576.3 |
| EGF | CAG GGA AGA TGA CCA CCA CT | AGC CAA CAA CAC AGT TGC AT | 140 bp | NM_001178130.3 |
| PGF | TGCCTTCAACAACGTGAGAG | AGGATCCGCATCCCTACTTT | 242 bp | NM_002632.6 |
| C3 | GGAGCAGTCAAGGTCTACGC | ACTTGATGGGGCTGATGAAC | 346 bp | NM_000064.4 |
| C5 | ACATTACGAGTGGTGCCAGAA | CCCTTTGGGGAGGTGGGTTA | 243 bp | NM_001735.3 |
| CCL2 | TCA AAC TGA AGC TCG CAC TCT | GGG GCA TTG ATT GCA TCT GG | 123 bp | NM_002982.4 |
| ESR1 | AAT TCA GAT AAT CGA CGC CAG | GTG TTT CAA CAT TCT CCC TCC TC | 345 bp | NM_000125.4 |
| ESR2 | TAG TGG TCC ATC GCC AGT TAT | GGG AGC CAC ACT TCA CCA T | 393 bp | NM_001437.3 |

## *In vivo* experiments; ovariectomy and sham surgery

Eye and serum samples for our investigations were obtained by our partners from the Department of Experimental Surgery, Charité – Universitätsmedizin Berlin, following Charité's C3R concept (where the experiments were approved by the local government authorities (Landesamt für Gesundheit und Soziales, LAGeSo, Berlin) under G0118/21). We used samples from young and old female Lewis rats obtained from Janvier Labs (Le Genest-Saint-Isle, France). The animals had either a physiological estrus and hormonal profile or a surgically early estrous disruption induced in select donor animals through bilateral ovariectomy. Bilateral ovariectomy was performed at 8 months of age to model early-onset menopause. The control group underwent sham surgery at the same age to mimic surgical and anesthetic stress without ovary removal. The procedure was as follows: ovariectomy [38] was performed in 8 months old rats under isoflurane anesthesia following premedication with buprenorphine (0.02 mg/kg s.c.) and metamizole (100 mg/kg s.c.). After induction with 3.5% isoflurane in oxygen, the abdomen was shaved and disinfected with 70% ethanol. Eyes were protected with VITA POS ointment. Anesthesia was maintained with 1–1.5% isoflurane, adjusted based on respiration and circulation. A midline laparotomy was performed, the wound edges were secured, and the small intestine was repositioned. The ovary was mobilized, ligated (silk 6−0), and excised using spring scissors. The procedure was repeated bilaterally. The abdominal cavity was closed in two layers with absorbable suture material. Before skin closure, wound edges were treated with bupivacaine gel for pain relief, and the skin was sealed with Vetbond tissue glue. Surgery lasted ~20 minutes. Rats were placed in a prewarmed recovery cage and monitored. Postoperative analgesia included carprofen (5 mg/kg s.c., 2 days) and tramadol (20 mg/100 ml drinking water, 3 days).

The sham surgery followed the same protocol as ovariectomy but without ovary removal Table 3.

**Table 3. material and medication for surgeries.**

| Material | Source |
|---|---|
| BUPRENOVET MULTI | Elanco, Bad Homburg, Germany |
| Metamizole, Novaminsulfon-ratiopharm | Ratiopharm, Ulm, Germany |
| Isoflurane | CP-Pharma, Burgdorf, Germany |
| VITA POS eye ointment | URSAPHARM, Saarbrücken, Germany |
| Arterial clamp, Mosquito curved 12.5 cm | Aesculap AG, Tuttlingen, Germany |
| Mass ligature, black silk 6−0 | Resorba, Nuremberg, Germany |
| Absorbable, atraumatic suture material, PDS II | Ethicon, Johnson & Johnson MedTech, Norderstedt, Germany |
| Bupivacaine gel 1% | Charité Apotheke, Berlin, Germany |
| Vetbond tissue adhesive | 3M, St. Paul, USA |
| Carposol | CP-Pharma, Burgdorf, Germany |
| Tramal Drops | Grünenthal, Stolberg, Germany |

### *Ex vivo* experiments; hormone profile

To analyze the hormone profile, blood samples were collected *ex vivo* and centrifuged at 3000 × g for 10 minutes. Serum was separated by pipetting the supernatant and stored at −20°C until analysis. The analyses were conducted using ELISA and LC-MS/MS techniques at Dresden Lab Service GmbH (Dresden, Germany).

### *Ex vivo* experiments; PCR analyses

For qPCR analyses of estrogen receptors and inflammatory markers, the RPE was isolated from enucleated eyes [39]. We made a circular incision to dissect the cornea before removing the lens and vitreous. The eyecups were flattened using four to five radial incisions extending from the peripheral fundus toward the optic nerve. The optic nerve was severed, allowing removal of the retina by separation of the retina and from RPE. The remaining part, which includes the RPE, choroid, and sclera, was then snap-frozen in liquid nitrogen and stored at −80°C until RNA isolation. Total RNA was extracted from rat RPE using the RNeasy Plus Mini Kit, according to the manufacturer's user manual (Table 4). The cDNA was synthesized with the QuantiTect Reverse Transcription Kit. Primers for *ex vivo* qPCR were obtained from Eurofins Genomics (Table 5). Gene expression was assessed using the Quantinova SYBR Green PCR Kit on a RotorGene real time PCR system. The housekeeping gene (GAPDH) showed consistent expression at 13–15 cycles. The results of the target genes were analyzed as described in the in vitro section.

### *Ex vivo* experiments; immunohistochemistry in sagittal sections and flatmount samples

The eyes for paraffin-embedded sagittal sections were fixed after the removal for 48 hours in 4% paraformaldehyde (PFA). Following this, the eyes were transferred into 70% ethanol and stored at 4°C until the embedding process. The embedding, sectioning and hematoxylin-eosin staining were performed by our technical assistant Ms. Oberländer and

**Table 4. Material information for *ex vivo* PCR analyses.**

| Material | Source | Cat.No |
|---|---|---|
| RNeasy Plus Mini Kit | Qiagen, Hilden, Germany | 74136 |
| QuantiTect Reverse Transcription Kit | Qiagen, Hilden, Germany | 205314 |
| SYBR Green PCR Kit | Quigen, Hilden, Germany | 208056 |

**Table 5. Primers used for *ex vivo* PCR analyses.**

| Gene | Forward | Reverse | Product length | Template |
|------|---------|---------|----------------|----------|
| Ccl2 | TGTAGCATCCACGTGCTGTC | CTTGAGCTTGGTGACAAATACTACA | 165 bp | NM_031530.1 |
| C3 | ACT CCA AAC CTT TCC ACC CC | ACCTCATCTGAGCCTGACTTG | 175 bp | NM_016994.2 |
| Pgf | CAGTGAACGGGACACCCATC | AGTTGTTCCCAGCAGACAGG | 194 bp | NM_053595.2 |
| Il1b | TAGCAGCTTTCGACAGTGAGGAG | CCTTCCTGAAGCTCTTGTCGAG | 156 bp | NM_031512.2 |
| Esr1 | TTG CTC TTG GAC AGG AAT CAA G | TCG GTG GAT GTG GTC CTT CT | 210 bp | NM_012689.2 |
| Esr2 | TCG TTC TGG ACA GGG ATG AG | GCA GAA GCC AAG GGG TAC AT | 169 bp | NM_012754.3 |
| Cxcl2 | ACC ATC AGG GTA CAG GGG TT | CAC CGT CAA GCT CTG GAT GT | 101 bp | NM_053647.2 |
| Gapdh | TTGTGCAGTGCCAGCCTC | TTGTCACAAGAGAAGGCAGC | 106 bp | NM_017008.4 |

the Central Biobank Charité (ZeBanC). Paraffin-embedded eyes were deparaffinized in an oven for 20 minutes following sequential rehydration: 20 minutes in ROTICLEAR, followed by two rounds of 10 minutes in isopropanol, and 5 minutes each in 96%, 90%, 70%, and 50% ethanol. For antigen unmasking, the samples were incubated in ethylenediaminetetraacetic acid (EDTA) buffer (pH 9.0) for senescence staining (20 minutes in a steamer), or with proteinase K for the ERß staining (7 minutes at room temperature). After washing three times with Tris-buffered saline (TBS) for 5 minutes each, the samples were permeabilized in 0.5% Triton-X 100 for 15 minutes. A one-hour blocking step with 5% serum albumin (BSA) was performed before adding the primary antibodies (Table 6 A and 6B). The next day, the primary antibodies were removed, and the samples were washed again with TBS. Secondary antibodies (Table 7 A and 7B) were added along with 3μM 4',6-diamidino-2-phenylindole (DAPI) and incubated for one hour at room temperature. After a final wash, coverslips were applied, and slides were stored in the dark at 4°C until microscopy with a ZEISS Axio Imager.M2 fluorescence microscope (Carl Zeiss Microscopy Germany, Jena) at 200- or 400-times magnification.

To prepare flatmount samples, we used a method that has been published earlier [39–41]. In short, the eyes were fixed in 4% PFA for 13 minutes and then washed in Dulbecco's phosphate-buffered saline (DPBS) for 15 minutes. The subsequent preparation procedure was similar to that used for qPCR analyses; by a circular incision the cornea was dissected and the lens and vitreous removed, the eyecups were flattened using four to five radial incisions from the peripheral fundus toward the optic nerve, the optic nerve was severed and the retina removed. The remaining

**Table 6. primary antibodies for immunohistochemical staining Supplemental Information available zenodo; "S6 Tabl_Antibody_Registry_beta_ID_Tab6_Tab7".**

| | Primary antibody/Lectin | Concentration |
|---|--------------------------|---------------|
| A | CDKN2B/CDKN2A/p16 Monoclonal Antibody (C-7)-Alexa Fluor 488, mouse IgG1 | 1/100 |
| | p21 Monoclonal Antibody, rabbit IgG | 1/1000 |
| B | ESR2 phospho S105 Polyclonal Antibody, rabbit IgG | 1/250 |
| C | Acti-stain 555 Fluorescent Phalloidin (Lectin from *Amanita phalloides*) | 1/500 |
| | Anti-Iba1 Monoclonal Antibody (EPR16589), rabbit IgG | 1/500 |

**Table 7. Secondary antibodies for immunohistochemical staining Supplemental Information available zenodo; "S6_Antibody_Registry_beta_ID_Tab6_Tab7".**

| | Secondary antibody | Concentration |
|---|--------------------|---------------|
| A/B | Donkey anti-rabbit IgG (H+L)-Cy3 AffiniPure, Polyclonal | 1/100 |
| C | Donkey IgG anti-rabbit IgG (H+L)-Alexa Fluor 488, Polyclonal | 1/500 |

flatmount, containing the RPE, choroid, and sclera, were permeabilized by incubating them in 5% Triton-X 100 at 4°C for 12 hours, preparing them for staining the following day. To block nonspecific binding, samples were incubated for 24 hours at 4°C with 15% BSA before the primary antibodies (Table 6 C) were applied. After 3 days of incubation at 4°C, the flatmounts were washed 3 times (10 minutes each on the shaker) to apply the secondary antibody (Table 7 C), with which the samples were incubated for 1 hour at room temperature on a shaker. Finally, three additional washes were performed before the samples were mounted and stored in the dark at 4°C until scanning with the confocal microscope Leica SPE at 200 magnification. The RPE and Iba1⁺ cells were analyzed in projected z-stack scans of the peripheral and central RPE-flatmounts by using the "Cell Count - Cellpose" recipe in AIVIA version 14.1.0 (Leica Microsystems GmbH, Wetzlar, Germany).

### Data analysis

For the statistical analyses, the data were tested for normal distribution using the Shapiro-Wilk test. Afterward, normally distributed datasets were compared using an unpaired *t*-test, and nonparametric datasets were compared using the Mann-Whitney test. The results are shown as a scatter plot with bars as mean with SEM or as box plots with mean and whiskers from minimum to maximum. Values of $*P < 0.05$, $**P < 0.01$, $***P < 0.001$, $****P < 0.0001$ were marked as statistically significant.

## Results

### The phenotype of structural changes in the aged and aged retina after early menopause

To evaluate effects of reduced estradiol levels onto the retina, we used either aged (22 months old) or ovariectomized (22 months old) Lewis rats (Fig 1A) as models for estradiol deficiency. Serum analyses revealed a broader spectrum of alterations of the hormone levels in the deficit models (Figs 1B and 1C). As controls served 6 months old Lewis rats. In comparison with 6 months old rats, the deficit models showed decreased levels of estradiol as expected but also significant effects on other steroid hormones. Progesterone and corticosterone were increased in the estradiol deficit models. Furthermore, we observed an increase in DHEA-S under estradiol deficiency conditions (Fig 1B). When comparing 22 months old sham treated with ovariectomized rats (Fig 1C), we found no significant differences in most hormones of the investigated panel, suggesting that the physiological decline in estradiol production and an induced premature reduction in estradiol production result in a similar hormone profile at this advanced age. However, the serum levels of DHEA-S were reduced in the ovariectomized rats compared with the sham treated rats.

Following the hypothesis that reduced estradiol levels might contribute to retinal degenerative alterations, we investigated the retina structure in 6 months, 22 months and 22 months ovariectomized rats (Fig 2A). For that structural analysis, we quantified the cell density of nuclei in the ganglion cell layer (GCL) and the density of the outer nuclear layers (ONL) in HE stained sagittal retinal sections. Indeed, the hormone status is associated with changes in retinal structure. As the 22 months sham treated rats showed no differences in the GCL compared with 6 months old rats, the comparison with ovariectomized rats exhibited a strong reduction in the ganglion cell nuclei density (Fig 2B). Furthermore, we found a clear manifestation of photoreceptor degeneration (Fig 2C). Here, compared with 6 months old rats, sham treated 22 months old rats had a significant reduced number of nuclei in the ONL, while ovariectomized rats showed an even more pronounced reduction in the number of nuclei.

Cellular senescence is a major mechanism of degeneration in ageing. To assess tissue ageing, we examined the expression of senescence-associated markers p16 and p21 in retinal sections from 6 months control rats and 22–24 months old rats (Fig 3). The staining occurred in retinas from rats at the age of 22 and 24 months. The controls did not show any reliable staining for p16 and p21. In old rats, we found p16 positive nuclei in the GCL and abundant positive nuclei in the INL and ONL. The staining for p21 had comparable results with positive nuclei in the GCL, INL but not in the ONL.

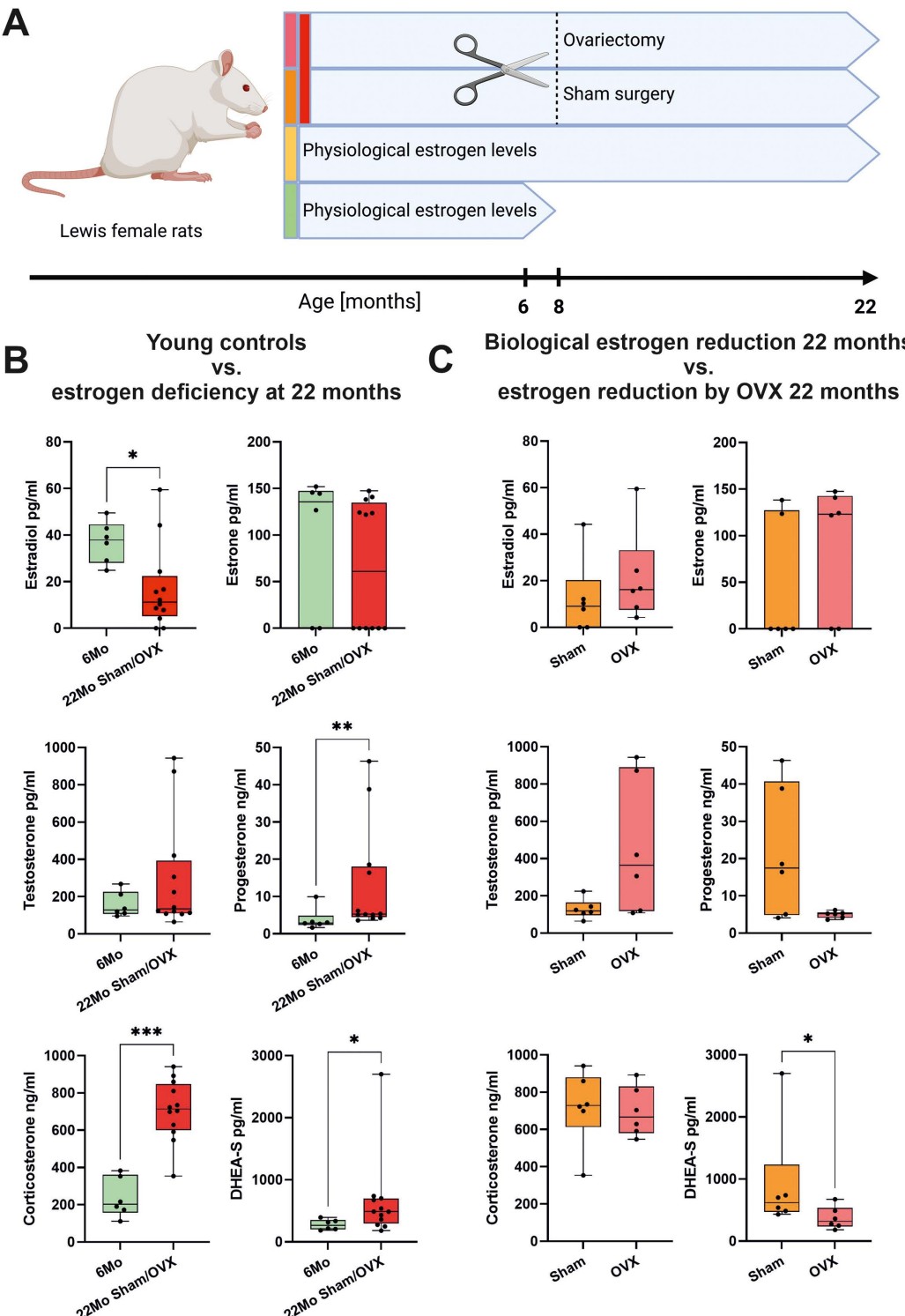

**Fig 1. Rat model of estradiol deficit. A)** Experimental Groups, created with BioRender.com **B)** Serum concentration of hormones depending on age (green = 6 months, red = 22 months). **C)** Serum concentration of hormones depending on surgical early menopause or during biological ageing (orange = Sham, red = OVX). Mann-Whitney tests were performed and the results are shown as boxes and whiskers (number of serum samples: 6Mo n = 6, 22Mo n = 12, Sham n = 6, OVX n = 6, *P < 0.05, **P < 0.01, ***P < 0.001).

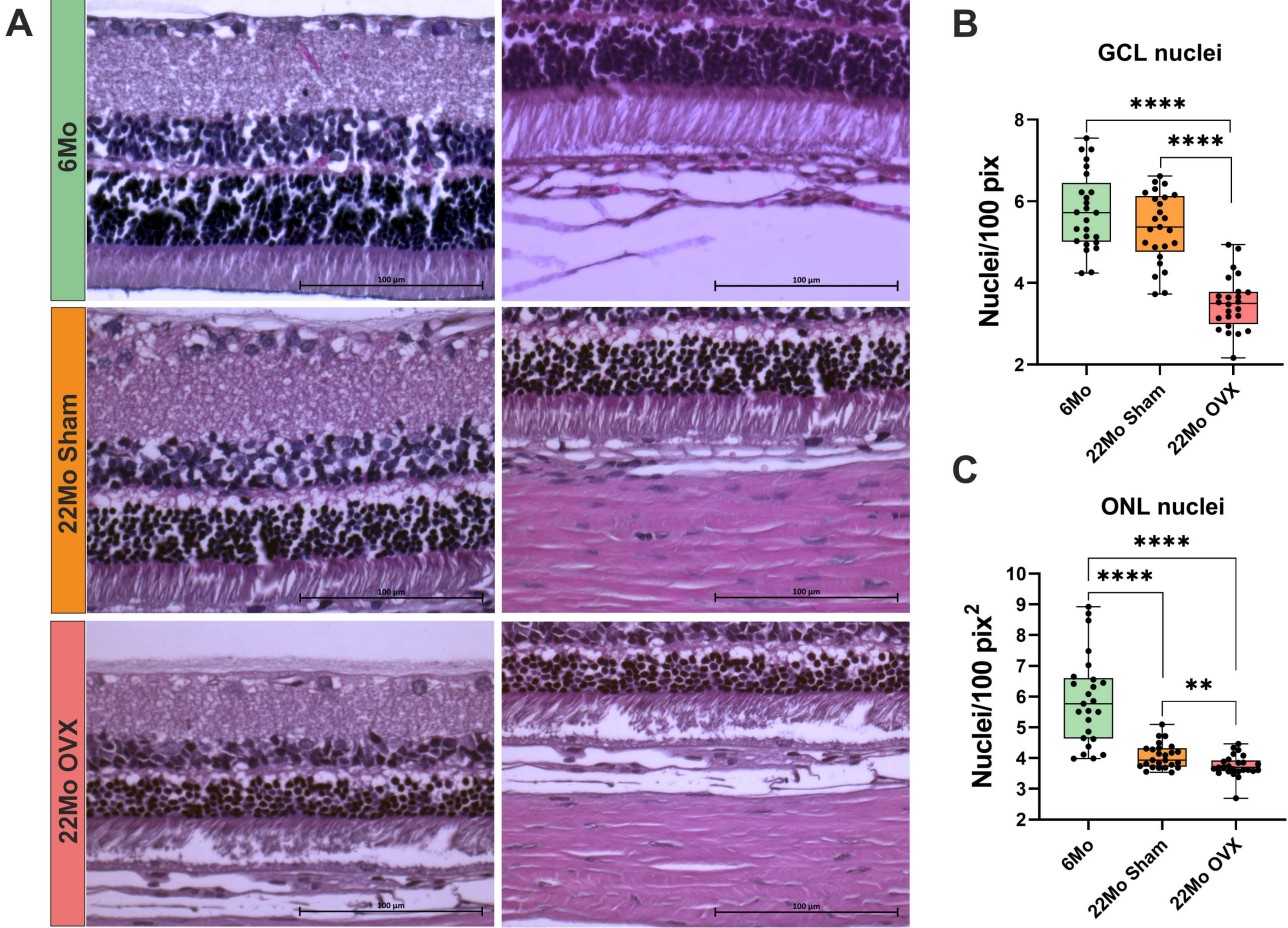

**Fig 2. Degeneration and cell loss. A)** HE stained sagittal sections (400x magnification, scale bar = 100μm); left column shows the inner retina, right column shows the RPE. **B)** Cell loss per 100pixel GCL, **C)** Cell loss per 100pix² ONL (green = 6 months, orange = 22 months Sham, red = 22 months OVX). Unpaired *t*-tests were performed and the results are shown as boxes and whiskers (n = 25 measuring points per group; 5 eyes from 5 different animals with 5 measuring points per eye, ***P* < 0.01, *****P* < 0.0001). Raw data available zenodo; "Fig2_HE_samples_cell_counting".

As AMD is characterized by a loss of RPE cells, which subsequently leads to photoreceptor degeneration, we examined the RPE cell density in RPE-flatmount preparations in correlation with the hormone status. The RPE cell borders were stained by phalloidin (Fig 4A). The cell density was quantified using an AI-based algorithm. As a measure of the cell density, we used the mean of size/shape parameters of RPE cells under the hypothesis that the RPE cell loss leads to larger cells to maintain the confluent monolayer of the same area or to the appearance of condensed small apoptotic cells. In all measures (mean size, min size and max size, StDev) the two estradiol deficit models showed significant higher numbers in increased and condensed cell sizes compared to the 6 months control (Figs 4B and 4C). However, in comparison between the models, sham treated rats and ovariectomized rats, we found no differences.

## Pathways that connect decreasing systemic estradiol levels with retinal degeneration

The RPE forms the outer blood-retina barrier making it one of the eyes' first tissues to respond to systemic estradiol fluctuations. Additionally, RPE dysfunction plays a central role in the pathogenesis of AMD. Thus, we investigated the estradiol receptor expression in RPE cells. At the level of cell physiology, we investigated RPE cells of the cell line ARPE-19

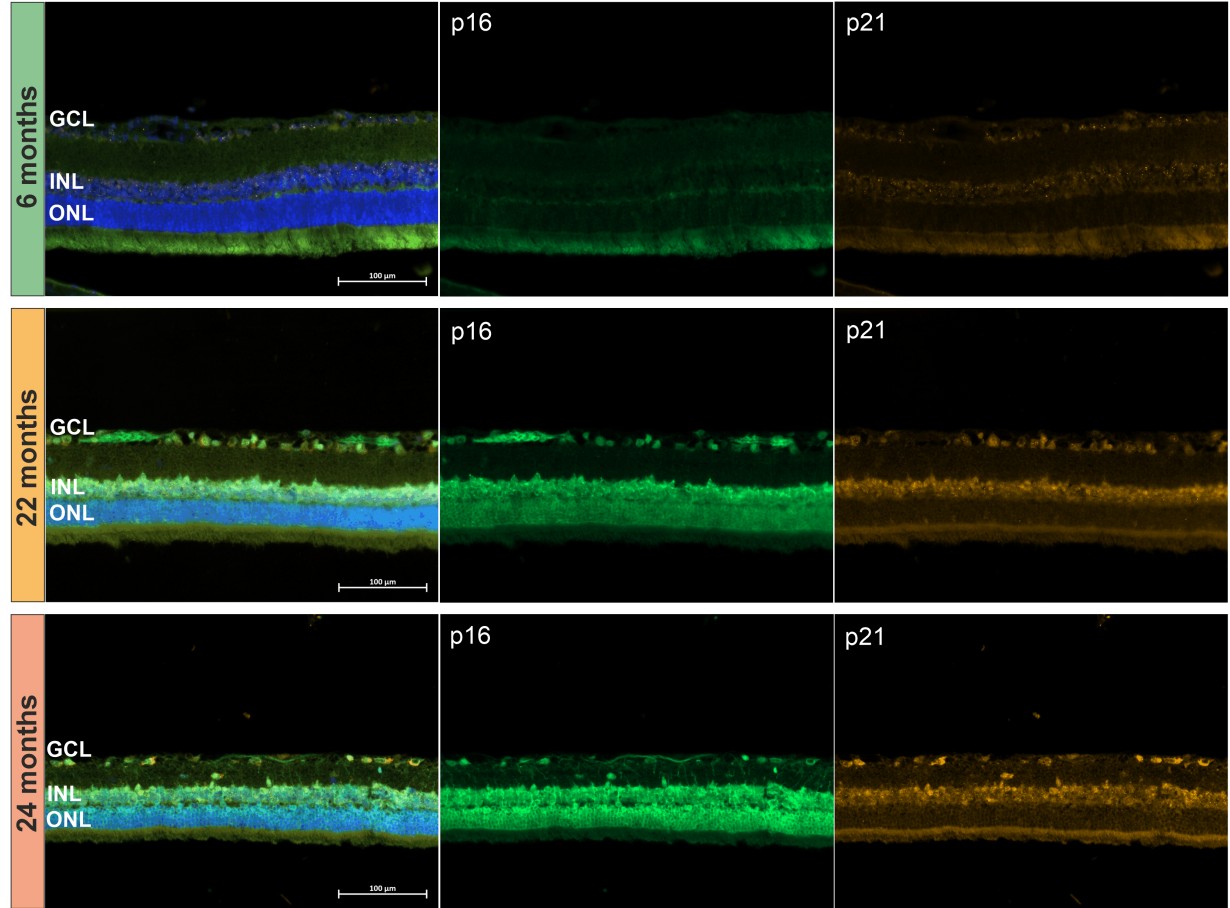

**Fig 3. Marker of cellular senescence.** IHC staining of p16 (green) and p21 (orange) in sagittal sections of rat retina (200x magnification, scale bar = 100μm). Raw data available zenodo; "Fig3_samples_p16p21_staining".

in a semi-confluent state to mimic a degenerative status. RT-PCR confirmed expression of both estradiol receptors ERα and ERβ in ARPE-19 cells (Fig 5A). To assess the functional ER activity, we stimulated ARPE-19 cells with estradiol. For stimulation, we used the concentration of 4μM that was in the range to change gene expression in ARPE-19 cells [42]. In binding assays of the isolated ERα and ERβ 1μM is in early saturation range [43] so that diffusion of extracellular estradiol to the ERs in the APRE-19 cells might need higher concentrations. Furthermore, the biologically effective estradiol concentrations result from co-stimulation with other hormones [42]. We measured differential expression of genes related to pro-inflammatory responses to estradiol stimulation by qPCR: complement factors (*C3*, *C5*, *CFH*), cytokines (*CCL2, IL-1β, IL-8*), members of the VEGF signaling system (*VEGF-A*, *PGF*) and RPEs' survival factor *EGF*. Stimulation of ARPE-19 with estradiol showed among the genes of this panel only an effect for *PGF* and *IL-1β* (Fig 5B). Here, estradiol significantly increased *IL-1β* expression and significantly decreased *PGF* expression. Both results showed the functional expression of ERs. The expression of estradiol receptor could be confirmed *in vivo*. qPCR showed the expression of ESR1 (ERα) and ESR2 (ERβ) in the retina at the mRNA level (Fig 5C). The ER expression changed under low estradiol conditions. We found that specifically the expression of *ESR2* was reduced in retinas of 22 months old ovariectomized rats. These findings suggest that estradiol depletion primarily affects ERβ activity in the RPE, potentially contributing to early AMD pathogenesis. By means of immunohistochemistry, we found ERβ positive signals in the RPE, in cells of the INL and the

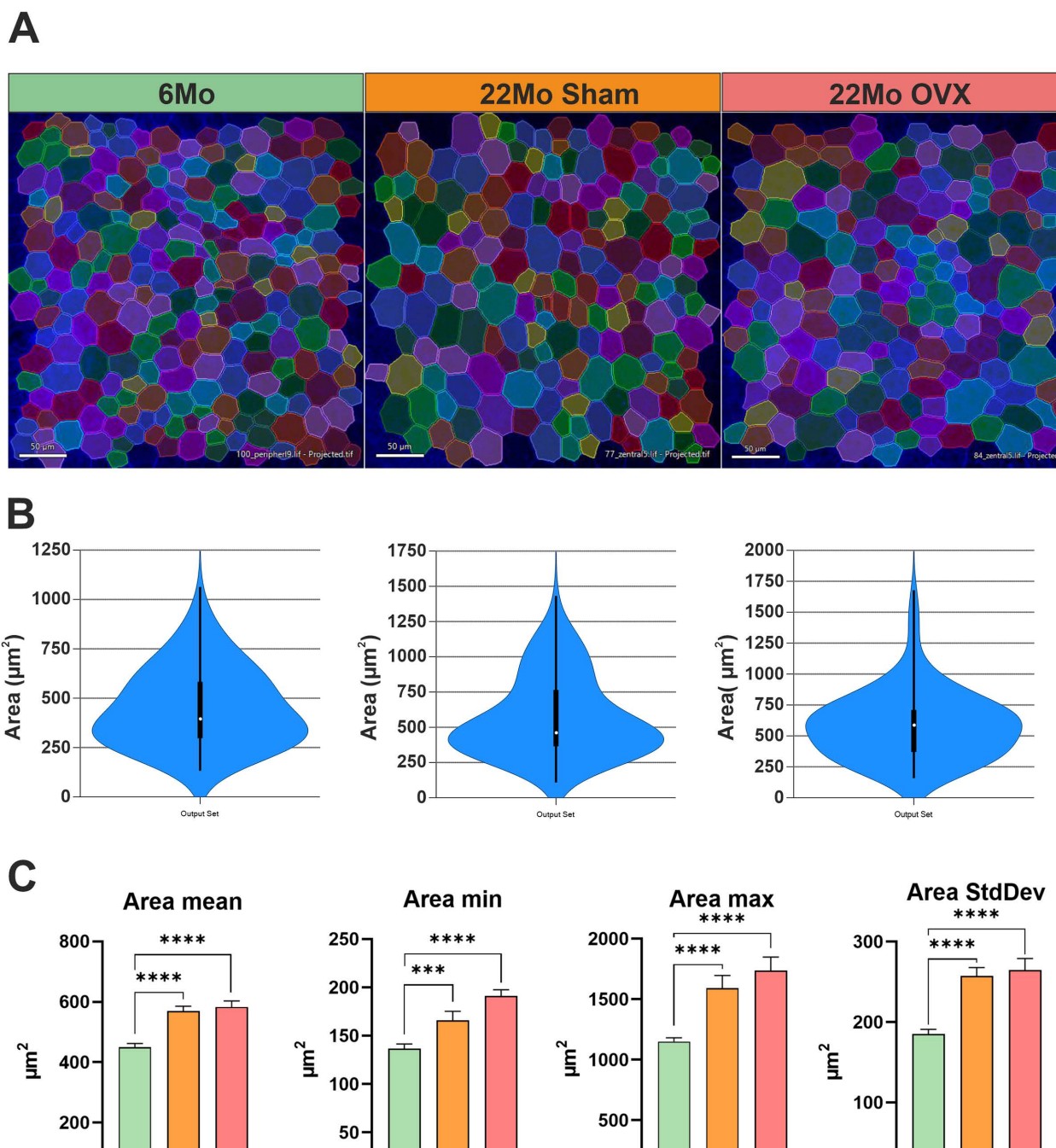

**Fig 4. Structural changes in the RPE-layer. A)** Segmentation of phalloidin-stained RPE cells (200x magnification, scale bar = 50µm). Raw data available zenodo; "S2_raw_files_phalloidin_staining_Fig4" **B)** Segmentation from row A as violin plots. **C)** Changes in the morphology of the RPE-layer analyzed with the AI-supported software AIVIA (green = 6 months, orange = 22 months Sham, red = 22 months OVX). The raw data by AIVIA were compared using Mann-Whitney tests and the results are shown in graphs as mean±SEM (6Mo; n = 58 scans out of 5 animals, 22Mo Sham; n = 46 scans out of 8 animals and 22Mo OVX; n = 80 scans out of 11 animals, ****$P$ < 0.0001, ***$P$ < 0.001).

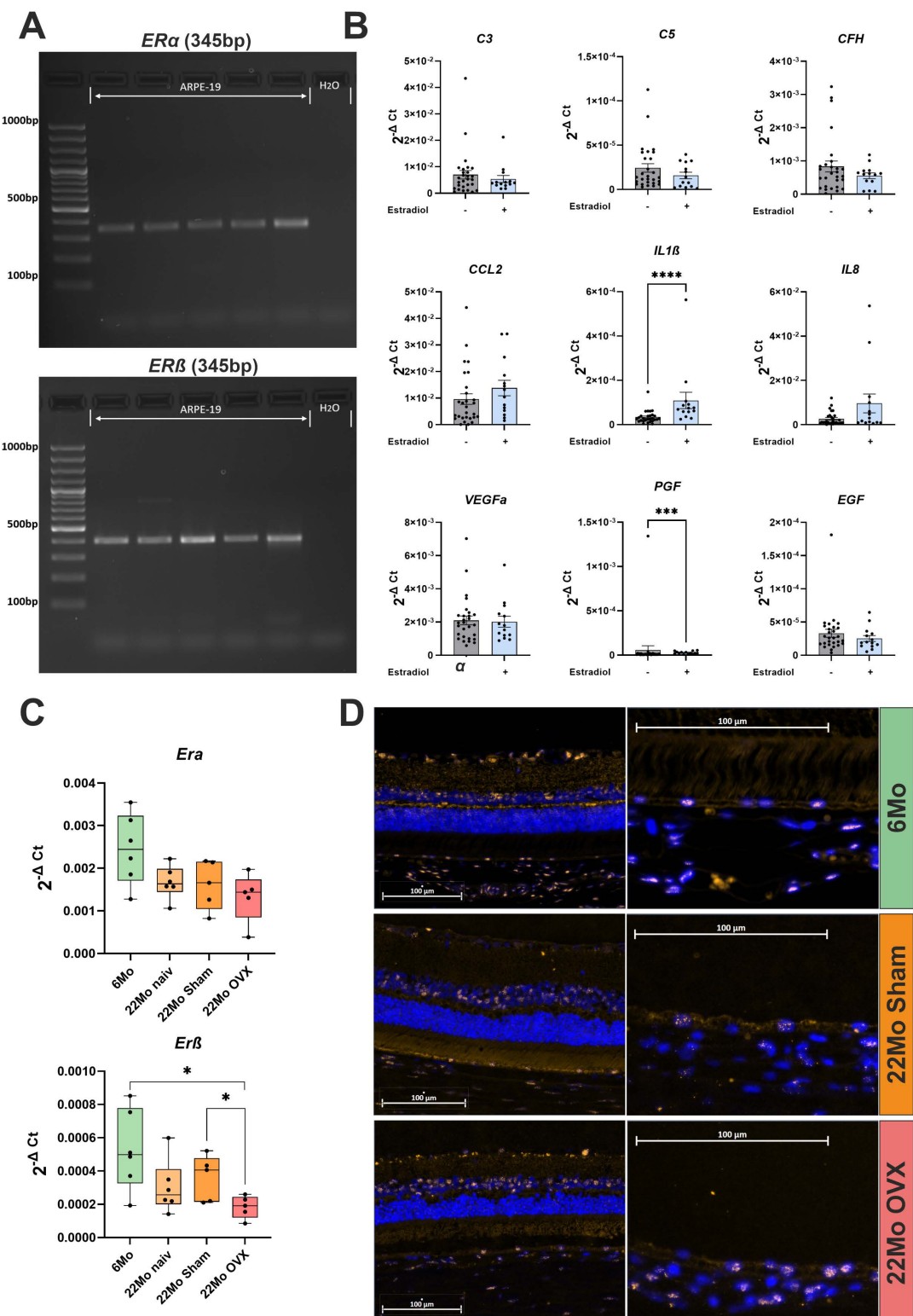

**Fig 5. Estrogen receptors in the RPE. A)** Ethidium bromide gel of PCR amplicons of the receptor ERα and ERβ in ARPE-19 cells. Raw data available zenodo; "S1_raw_images_Fig5" B) Gene expression change depending on estrogen, in ARPE-19 cells (gray=ARPE-19 without estradiol; n=28, blue=ARPE-19+4µM estradiol; n=14), Mann-Whitney tests were performed and the results are presented in the diagram as mean±SEM (***$P<0.001$,

****$P < 0.0001$). C) qPCR detection of the receptors ERα and ERβ in rat RPE samples (green = 6 months; n = 6, light orange = 22 months naïve; n = 6, orange = 22 months Sham; n = 5, red = 22 months OVX; n = 5), analyzed by using t-tests and presented as boxes and whiskers (*$P < 0.05$). **D)** IHC staining of ERß (orange), left in the inner retina (200x magnification, scale bar = 100μm), right in the RPE (400x magnification, scale bar = 50μm). Raw data available zenodo; "Fig5_samples_ERß_staining".

ganglion cells (Fig 5D). In summary, we can assume that the first effects of estradiol deficits evolve by ER deactivation in the RPE.

Chronic cellular inflammation in the outer retina is a key factor in the etiology of AMD. If low systemic estradiol levels are associated with an increased risk of AMD, we hypothesized that estradiol must exert anti-inflammatory effects in the retina. To test this hypothesis, we conducted gene expression analysis in ARPE-19 cells under pro-inflammatory conditions comparable to that of AMD (Fig 6A). Specifically, all estradiol stimulation experiments were performed in the presence of 0.6nM TNFα, a cytokine released by monocyte-derived macrophages that induces pro-inflammatory changes in RPE cells [33,34] using the same panel of inflammation-related genes as before. Under these conditions, estradiol significantly reduced the expression of the complement factor *C5* and increased the expression of *PGF* whereas the expression of the other genes remained unchanged.

To obtain more insights from *in vivo* models, we performed qPCR-based gene expression analysis (Fig 6B) and quantified the invasion of mononuclear phagocytes (Iba1+ cells) in RPE/choroid flatmounts of the outer retina (Figs 6C – 6E)). Indeed, we found an increased level of *Ccl2* expression in the 22 months old ovariectomized rats. The quantification of Iba1+ cells occurred using an AI-based software and aimed the total number of cells and the number of cells in an amoeboid morphology (Figs 6D and 6E), where the latter indicated the status of activated cells. While the total number of Iba1+ cells was similar across all groups, we observed a significant increase in amoeboid-shaped (activated) Iba1+ cells in the 22 months old ovariectomized rats. Thus, estradiol deficit potentially fosters but unlikely causes cellular inflammation in the outer retina.

## Discussion

Clinical data suggest a strong correlation between serum estradiol levels and AMD risk. Our study proposes a mechanistic explanation for how estradiol deficiency may contribute to AMD pathogenesis. To investigate the impact of aging and estradiol deficiency on retinal degeneration, we used aged Lewis rats and ovariectomized Lewis rats as models. This study is the first to examine the long-term effects of aging and ovariectomy on the retina in correlation with serum gonadal hormone levels in female rats. Both biological and surgically induced estradiol deficiency resulted in the loss of ganglion cells, photoreceptors, and RPE cells in the retina. In ganglion cells, cells with nuclei in the INL, and RPE cells we found the expression of ERß. We found mild estradiol impact on cellular inflammation and that the ERß positive cells are those that expose senescent markers with age. Hormone analysis in the rat models revealed a primary reduction in estradiol levels, accompanied by compensatory changes in other hormones, including increased progesterone, corticosterone and DHEA-S levels. Thus, our study does not show that estradiol deficiency cause AMD but increases the risk by promoting cellular inflammation, senescence and retinal degeneration in ganglion, RPE and secondary to RPE also in photoreceptor cells. However, the early menopause risk effect potentially depends on non-estradiol effects on photoreceptors.

To study the AMD risk factors of age and estradiol deficiency, we used aged Lewis rats that underwent an ovariectomy or sham surgery at the age of 8 months, to compare them and determine differences compared to 6 months old rats. The aged rats show the expected decrease in estradiol serum levels but also profound changes in other hormones compared to young rats: an increase in the levels of progesterone, corticosterone, and DHEA-S. Interestingly, the only significant difference between aged rats after sham or ovariectomy was significantly reduced level of DHEA-S in the animals after ovariectomy. In this way, our study differs from the ones of the existing literature [44–50] in which the observational periods

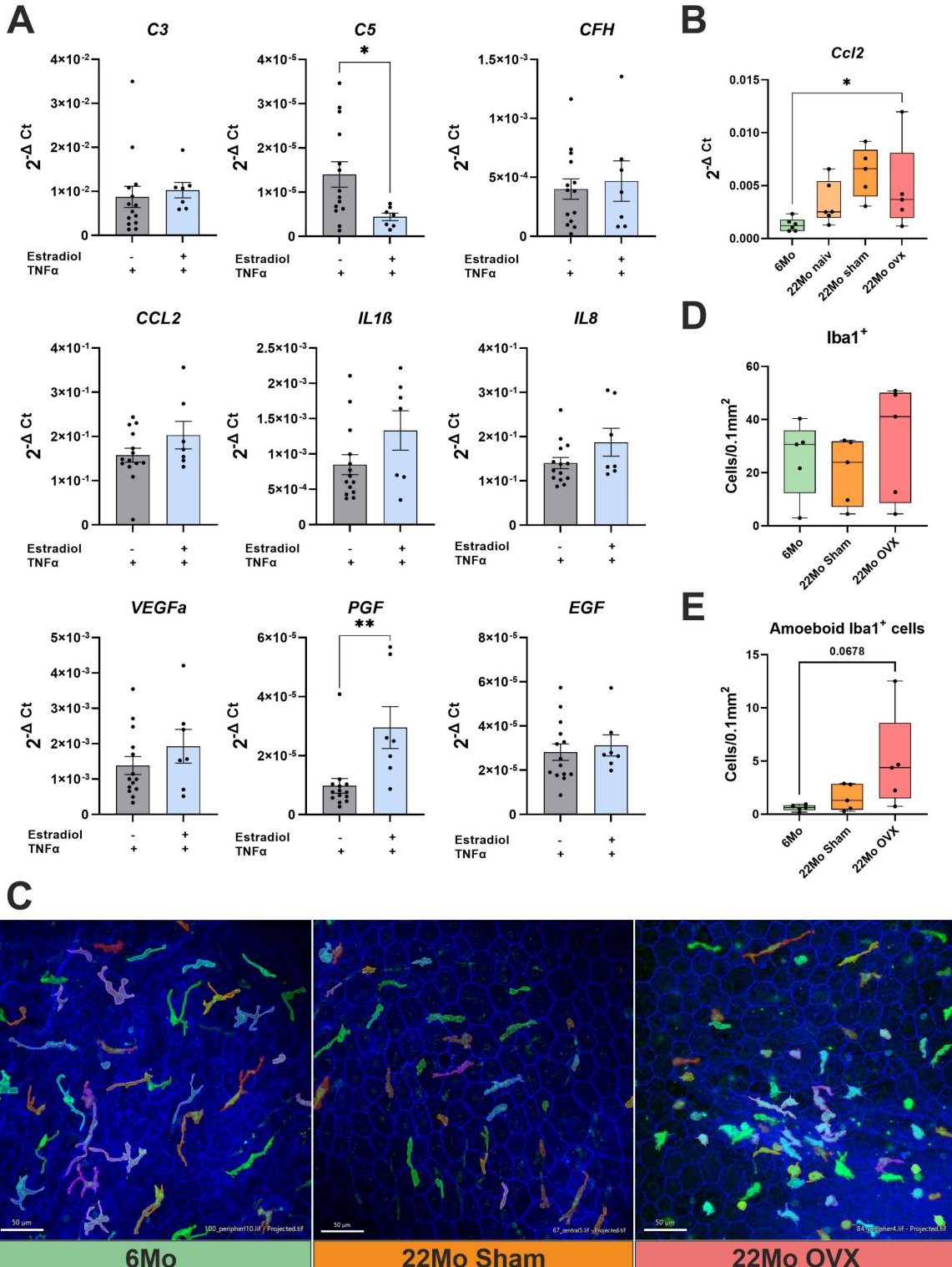

**Fig 6. Estrogen impact on inflammatory processes in the RPE. A)** Gene expression changes of ARPE-19 cells under inflammatory (TNFα) conditions (grey = without estradiol; n = 14, blue = + 4μM estradiol; n = 7), analyzed by Mann-Whitney tests and presented in the diagram as mean ± SEM (*P < 0.05, **P < 0.01). **B)** Ccl2 expression level of the rat RPE depending on the estrous cycle (green = 6 months; n = 6, light orange = 22 months naïve;

n = 6, orange = 22 months Sham; n = 5, red = 22 months OVX; n = 5), evaluated by Mann-Whitney tests and shown as boxes and whiskers (*P < 0.05). **C)** Images of the Iba1 analysis with the AI-supported software AIVIA (200x magnification, scale bar = 50µm). Raw data available zenodo; "S3_raw_files_Iba1_staining_Fig6". **D)** AI-supported analysis of Iba1$^+$ cells (n = 5 scans from 5 different animals per group). **E)** Counting of amoeboid Iba1$^+$ cells (with scans from 5 animals per group with the following number of scans: 6 months; n = 53, 22 months Sham; n = 25, 22 months OVX; n = 36).

following ovariectomy were significantly shorter, lasting only a few weeks. As a result, those studies primarily focused on the acute effects of estradiol deficiency, which have limited relevance for age-related macular degeneration (AMD). Furthermore, none of the studies evaluated the resulting hormone status so that the authors could only discuss estradiol deficits as cause for their observations without knowing the role of side effects. Another interesting observation was that at the age of 22 months, the two estradiol deficiency models showed almost no differences in their hormone status, indicating that biological aging resulted in a similar hormonal status as that of ovariectomy. However, DHEA-S levels were lower in the ovariectomized group. The reason for this remains unclear, but this finding underscores that ovariectomy differs from biological estradiol deficiency and does not exclusively reflect estradiol-related effects.

One major observation in the serum steroid hormone analysis was an increased progesterone level in aged rats compared to that of the young ones. Like estradiol is known in many pre-clinical studies being protective against various types of retinal damage [45,48,51–54] progesterone displays the same properties [55–60]. DHEA-S was also increased the estradiol deficiency models in comparison with the 6 months control. Also, DHEA-S is known as a protective factor in various neurodegenerative diseases [61–64]. Thus, the upregulation of progesterone and DHEA-S might reflect a compensatory response to estradiol deficiency. When evaluating degenerative changes, we found a reduction in ganglion cells only for the ovariectomy group compared to young rats. Since the hormone profile of this group differs just by a significantly reduced level of DHEA-S compared to age-matched rats after sham surgery, this finding may support the hypothetically compensatory and protective role of DHEA-S.

The estradiol deficiency that was found in both aged groups, had concrete structural effects on the retina. We found a loss of photoreceptor cells and RPE cells. We further found cellular senescence in estradiol deficiency-induced retinal degeneration. We analyzed p16 and p21 expression in retinas of aged rats that are cumulative senescence markers: p16 (cyclin-dependent kinase inhibitor 2A) and p21 (CDK-inhibitor 1) [65,66]. Both are involved in the cell cycle control and their activities depend strongly on the context: tumorigenesis, development or ageing [65,66]. As the 6 months control showed no specific staining, we found abundant p16 and p21 expression in aged rats. P16 was detectable in all nuclear layers, while p21 occurred just in ganglion cells and cells of the INL, but not in photoreceptors, which also showed no expression of the ERß. As we found a similar reduced level of serum estradiol and photoreceptor degeneration in ovariectomized rats and biologically aged animals compared to young rats, we suggest specific effects in the early menopause situation that do not depend on the reduction of systemic estradiol.

Grossly, this pattern seems to correlate with the estradiol receptor (ERs) expression. RT-PCR confirmed the expression of ERα and ERβ in RPE cells *in vitro*, and functional analysis revealed estradiol-induced changes in gene expression. These observations are supported by other studies also at the protein and functional level in the same RPE cell line but also in *in vivo* models [53,67–70]. qPCR showed in the *in vivo* retina the expression of ERα and ERβ. For further analysis of the ER protein expression and localization, we focused on ERβ because ERβ was found to exert protective effects in retinal [53,69–72] and brain injury [73,74] but also because ERβ was downregulated in the ovariectomized animals. The latter observation suggests a potential patho-physiological effect because estradiol deficiency potentially reduces neuroprotection by the molecule itself plus by reduced estradiol sensitivity of the cells. Immunohistochemistry shows that ERβ is expressed in the aged retina and in the aged ovariectomized retina in ganglion cells, cells with nuclei in the INL and in the RPE but not in the photoreceptors. This pattern corresponds with the existing literature [53,71,75,76]. The cells with nuclei in the INL might possibly be Müller cells [77] that are also known to express ERs. The degeneration of photoreceptors,

which lack ER expression and only show the p16 senescence marker, is likely secondary to RPE cell loss—a process that is also implicated in AMD pathogenesis.

Our study and also literature showed the expression of ERs in ganglion cells and that estradiol deficiency leads to ganglion cell loss. It is to mention that so far, the estradiol dependent loss of ganglion cells was reported as effects in observation periods up to 30 days [44,45,47,48,78,79]. In contrast, our data show ganglion cell loss after 22 months only in ovariectomized rat retinas, but not in age-matched sham-treated animals. These findings suggest that biological estradiol deficiency alone does not drive ganglion cell loss; rather, ovariectomy-related factors, such as reduced serum DHEA-S levels, may contribute to this phenomenon. In contrast, RPE cell loss, indicated by the increase of large-size and condensed RPE cells, occurred in both aged and ovariectomized rats, indicating that ER signaling plays a direct role in RPE survival. This conclusion would describe a potential specific pathway for the biological estradiol deficiency being a risk factor for AMD. Interestingly, the photoreceptor loss, that might be secondary to RPE loss, was stronger in the ovariectomized group than in sham-group. Also, here an estradiol-independent, potentially DHEA-S-dependent mechanism might lead to additional photoreceptor cells loss in the ovariectomy model. This observation is of significance for AMD. Two independent studies found an association of reduced DHEA-S serum levels and an increased prevalence for AMD [80,81]. It is possible that photoreceptors depend on the protective effects of DHEA-S and that reduction of DHEA-S in the ovariectomized rats additionally promotes the photoreceptor loss. Vice versa our data helps to understand the association between DHEA-S levels and AMD. However, the role of DHEA-S in the retina is so far unexplored.

To investigate how changes in gonadal hormone status contribute to retinal degeneration, we examined two potential mechanisms: local cellular inflammation and cellular senescence. It is well established that chronic cellular inflammation in the outer retina is an important mechanism in the pathology of AMD [35,82]. This inflammatory reaction centers around the invasion and activation of mononuclear phagocytes, an Iba1⁺ cell type that includes retinal microglia and monocytes from the blood stream. Due to their sensitivity to estradiol, monocytes influence the expression of several inflammatory factors depending on estradiol level [83]. The accumulation of mononuclear phagocytes requires the loss of barrier function by the RPE and a change of the RPEs' immunogenic phenotype towards an immune stimulatory phenotype [26]. One mechanism that helps monocytes to overcome the RPE barrier is the secretion of TNFα [33,34]. The observation that ERβ stimulation in monocytes increases the expression of TNFα [83] further supports a hypothesis for an ERβ – TNFα axis. Thus, we investigated *in vitro* the effects of estradiol on the RPEs' gene expression in the presence of TNFα. Among the AMD-relevant inflammatory genes, we found that estradiol reduced the expression of complement factor *C5* while increasing *PGF* expression. Thus, we must conclude that the estradiol deficiency would lead to an upregulation of *C5* and a downregulation of *PGF* expression. Increased activity of the complement system and accumulation of terminal complement complex in the AMD patients' retina occurs from early on and would explain the risk increasing effect of estradiol deficiency. PGF is a factor of tissue regeneration, and its downregulation would also be a factor that increases AMD risk. However, the estradiol effect on the gene expression is rather weak as only two genes appeared to be differently regulated. *In vivo* analysis showed in the RPE/choroid complex an upregulation of *Ccl2* (*Mcp1* = monocyte chemoattractant protein-1) [35,84,85] in AMD or AMD models, and exclusively in ovariectomized rats too. In this way low estradiol levels in the serum additionally support monocyte accumulation/activation in the ovariectomized rat retinas. Counting Iba⁺ cells in RPE flatmount preparations, we did not see a difference in the number of cells between all three animal groups. Already at the age of 6 months albino rats display a certain pro-inflammatory environment in the outer retina because the RPE expresses already the pro-inflammatory transcription factor FoxP3 [85]. However, again just the ovariectomized rats showed an increased number of amoeboid Iba1⁺ cells, a morphologic phenotype that indicates the activated status of mononuclear phagocytes. In summary, we saw a stimulatory impact on cellular inflammation in correlation with estradiol deficiency that supports other ongoing age-dependent mechanisms. Another reason for that observation might be the time point of investigation with 22 months. It is likely that at other timepoints in biological ageing or after ovariectomy might

                                                    

represent much stronger phases of cellular immune activity. Again, the upregulation of progesterone or DHEA-S downregulation might create confounders that attenuate the estradiol effect.

In summary, our findings indicate that biological aging and surgically induced early menopause result in a more complex pattern of gonadal hormone alterations than estradiol deficiency alone. This situation in combination with ageing leads to structural degenerative changes in the retina accompanied with estradiol deficit associated promotion of cellular inflammation and senescence. The estradiol-dependent promotion of cellular senescence seems to be more pronounced than that of cellular inflammation. However, rather than initiating a direct pathological cascade leading to AMD, estradiol deficiency likely acts as a modulator that increases susceptibility to retinal degeneration. Since early menopause is a recognized risk factor for AMD, the observed estradiol-dependent effects align with other known risk factors, such as complement system overactivation. Furthermore, the low penetrance of estradiol deficiency in AMD pathogenesis may be explained by compensatory upregulation of protective hormones, such as progesterone and DHEA-S, which may partially mitigate the degenerative effects of estradiol depletion. Concerning the photoreceptor degeneration and development of cellular senescence, the early menopause-dependent increase in AMD risk relies on other effects than low estradiol levels.

## Supporting information

**S1 Raw Images. Fig5. Original images of the gel electrophoresis.**
(PDF)

**S2 Raw Files. Phalloidin staining Fig4 Original images used for the RPE segmentation by AIVIA, shown in Figure 4.**
(PDF)

**S3 Raw Files. Iba1 staining Fig6 Original images used for the segmentation of Iba1 positive cells by AIVIA, shown in Figure 6.**
(PDF)

**S4 File Fig3 negative control. Negative control of the staining of Fig3.**
(TIF)

**S5 File. Fig3 peripheral. Staining of Fig3 in the peripheral retina.**
(TIF)

**S6 Table. Antibody Registry beta ID Tab6 Tab7. Additional information on the antibodies used, including ID from "Antibody Registry beta".**
(PDF)

## Acknowledgments

The authors express their gratitude to Prof. Dr. Clemens Kirschbaum from Dresden LABservice GmbH for performing the ELISA and LC-MS/MS analyses of hormones in rat serum.

We extend our special thanks to:
• Sergej Skosyrski for conducting the qPCR analyses of rat RPE samples,
• Yang Fang for her contributions to the immunohistochemical staining of the flatmount samples,
• Nikolaos Mylonas for his work on the immunohistochemical staining of the ERβ receptor in sagittal sections.

Additionally, we sincerely appreciate the Central Biobank Charité (ZeBanC) for their excellent collaboration in sectioning paraffin-embedded eyes and performing hematoxylin-eosin (HE) staining of sagittal sections.

## Author contributions

**Conceptualization:** Inga-Marie Pompös, Dietrich Polenz, Norbert Kociok, Olaf Strauß.

**Data curation:** Inga-Marie Pompös, Dietrich Polenz, Silvia Winkler.

**Formal analysis:** Inga-Marie Pompös, Norbert Kociok, Silvia Winkler, Olaf Strauß.

**Funding acquisition:** Inga-Marie Pompös, Olaf Strauß.

**Investigation:** Inga-Marie Pompös, Norbert Kociok, Olaf Strauß.

**Methodology:** Inga-Marie Pompös, Dietrich Polenz, Silvia Winkler.

**Project administration:** Inga-Marie Pompös, Dietrich Polenz, Olaf Strauß.

**Supervision:** Dietrich Polenz, Olaf Strauß.

**Validation:** Inga-Marie Pompös.

**Writing – original draft:** Inga-Marie Pompös, Olaf Strauß.

**Writing – review & editing:** Inga-Marie Pompös, Dietrich Polenz, Norbert Kociok, Olaf Strauß.

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
