## [Decision Letter · Decision Letter 0]

16 Sep 2025

The authors are requested to incorporate all the suggested revisions from the reviewers and thoroughly review and refine the manuscript before submission

We look forward to receiving your revised manuscript.

Kind regards,

Mohd Akbar Bhat, Ph.D.

Academic Editor

PLOS ONE

Journal Requirements:

4. Thank you for stating the following financial disclosure: [Deutsche Forschungsgemeinschaft DFG Dr. Werner Jackstädt Stiftung Stiftung Charité in cooperation with Einstein Foundation Berlin].

Reviewers' comments:

Reviewer's Responses to Questions

**Comments to the Author**

1. Is the manuscript technically sound, and do the data support the conclusions?

Reviewer #1: Yes

Reviewer #2: No

Reviewer #3: No

2. Has the statistical analysis been performed appropriately and rigorously?

Reviewer #1: Yes

Reviewer #2: Yes

Reviewer #3: No

3. Have the authors made all data underlying the findings in their manuscript fully available?

Reviewer #1: Yes

Reviewer #2: Yes

Reviewer #3: No

4. Is the manuscript presented in an intelligible fashion and written in standard English?

Reviewer #1: Yes

Reviewer #2: Yes

Reviewer #3: Yes

Reviewer #1: The manuscript entitled “Evaluation of risk-promoting effects for age-related macular degeneration by estradiol” describes how biological or ovariectomy-induced estradiol deficiency promotes mechanisms that lead to age-related macular degeneration. Further, the authors documented, the estradiol-dependent promotion of cellular senescence is more pronounced than that of cellular inflammation. Importantly, the authors documented, the estradiol deficiency probably acts as a modulator that increases susceptibility to retinal degeneration rather than initiating a direct pathological cascade leading to age-related macular degeneration. Overall, the manuscript is well written with few errors. I have one major concern, although the authors have shown nice images of hematoxylin and eosin staining, exhibiting photoreceptor degeneration and a reduction in the ganglion cell nuclei count in ovariectomized rats. I suggest performing immunohistochemical analysis employing rod and cone photoreceptor antibodies to further confirm the photoreceptor degeneration. A few minor comments are below.

1. Introduction section, line 96. Incorporate a reference related to hormone therapy during menopause.

2. Materials and methods section, line 143. Write the confluency of the cell culture as a percentage.

3. Materials and methods section, line 148. It should be “The primer sequences are provided in Table 1.”

4. Materials and methods section, line 150. Italicize the gene name throughout the manuscript, including the table.

5. Figure legends 2, 3, and 5. Write the magnification of the images captured.

Reviewer #2: This is a well-written and carefully executed study that investigates the contribution of estradiol deficiency to age-related macular degeneration (AMD) risk, with a focus on retinal degeneration, inflammation, and cellular senescence. The authors employed two rat models: naturally aged Lewis rats (22 months) to mimic biological estradiol decline and rats ovariectomized at 8 months, analyzed at 22 months, to model early menopause. Six-month-old rats served as controls. Methods included serum hormone analysis (ELISA, LC-MS/MS), qPCR for gene expression, immunohistochemistry for protein localization, AI-based image analysis for retinal cell density and inflammatory markers, and in vitro assays using ARPE-19 cells to assess ER function.

The study provides compelling evidence that estradiol deficiency does not directly cause AMD but promotes retinal degeneration, inflammation, and senescence, thereby increasing AMD risk. Importantly, the data also point toward broader hormonal influences, with DHEA-S potentially contributing to photoreceptor degeneration in early menopause. While the manuscript is strong, several limitations should be addressed to strengthen its conclusions:

Comment 1: The manuscript provides convincing structural evidence of retinal alterations; however, the absence of functional vision assessment (e.g., ERG) limits translational impact. Electrophysiological characterization would determine whether hormonal deficiencies result in measurable visual dysfunction, thereby enhancing clinical relevance to AMD progression.

Comment 2: The link between estradiol deficiency and p16/p21-driven senescence in ERβ-expressing retinal populations is compelling but requires deeper mechanistic insight. Single-cell RNA sequencing of retinal cell types (RGCs, INL cells, and RPE) from estradiol-deficient models could delineate cell type–specific senescence programs.

Comment 3: The study associates global hormone status (progesterone, corticosterone, DHEA-S, alongside estradiol) with retinal degeneration, inflammation, and senescence. However, the individual contributions of progesterone or corticosterone to the observed phenotype remain undefined. Clarifying their specific roles in retinal degeneration, inflammation, or senescence markers would strengthen the conclusions.

Comment 4: The use of 6-month-old rats as controls for both 22-month-old (aged) and ovariectomized 22-month-old rats may not adequately account for age-related changes independent of estradiol deficiency. An age-matched control group would better isolate the effects of estradiol loss from normal aging.

Comment 5: The authors propose that photoreceptor degeneration and cellular senescence in ovariectomized rats may occur through ERβ-independent mechanisms, potentially mediated by DHEA-S reduction. However, this interpretation lacks direct experimental support. Additional validation—such as DHEA-S supplementation studies or downstream pathway analyses—would provide mechanistic clarity and strengthen claims about non-estradiol hormonal influences on AMD risk.

Reviewer #3: In this manuscript, authors have utilized two different rat models to study estradiol deficiency-related changes that could potentially be associated with AMD. Some of the information provided in the abstract (line 47-48) was not seen in the manuscript, please remove that or else add the data in the manuscript or supplemental.

Authors are requested to address the following issues:

1. Methods section is poorly written: improper structure and missing information.

It should be structurally written method-wise along with #catalogue, brand information (wherever necessary).

Line 189-191: Briefly mention RNA isolation from RPE/choroid eye cups with suitable reference. State clearly if retina was removed or included in lysate preparation for RNA isolation/qPCR.

Line 218-219: RPE flat mount preparation is not similar to that of sample preparation for RNA isolation, as written by authors. State the flat mount procedure briefly and provide suitable reference for the method used. Not sure which method they have referred to, but I do not think 12-hour long permeabilization is recommended.

2. Statistical analysis: Why the authors did not use two-way ANOVA, since age is another parameter that needs to be addressed while comparing 6mo control vs 22mo naïve, Sham and OVX groups.

3. Use rat gene nomenclature throughout the manuscript.

4. Results section: Needs structure.

Fig. 2A: The ONL nuclei reduction seen in 22mo old rats vs 6mo control is probably due to aging (as reported earlier: INL, ONL thinning happens normally with aging), but what is surprising is the reduced thickness of the whole retina (especially INL) in 6mo old rats as compared to 22mo-old Sham rats. Explain.

Additionally, considering the significant expression of estrogen receptors in INL, I am not sure why INL quantification was not performed in these groups?

Fig2A (6mo, RPE): shows a huge white gap between IS/OS. Is this gap normal? Replace with a representative image.

Mention scale bar in legends.

5. Fig 2B: Since serum estradiol levels did not show significant change (in fact an increased pattern was seen in Fig 1C), what is the reason for the striking change in GCL and ONL nuclei? Explain in text.

6. Results: Fig 3, Does extreme left panel show merged image, please mention. Low quality images, replace with better resolution and higher magnification images.

In the abstract authors have mentioned more p16-positive photoreceptors were seen in 22mo OVX rats compared with Sham (line 46-48). Did the authors perform p16/p21-staining on 22mo OVX rats to see if estradiol deficiency induces more cellular senescence? If not, remove this section from abstract/discussion.

7. Line 294: ‘…RPE cell density in retinal flat mounts.’ Were these (Fig 4A) retinal or RPE flat mounts? State clearly.

8. Fig 4A: Add original phalloidin staining of these respective RPE regions above the segmentation panel.

The RPE flat mounts analyzed here (Fig 4A-B) represent which region: central, equatorial or peripheral? This is important since the RPE size/shape varies with different region. So, all three images compared here should be from same region. Add this important information in figure and legends.

9. Fig 4B: Increase text size of X-/Y-axes legends, too small to comment anything on this panel.

Apart from mean area, authors should add shape parameters like solidity, extent/eccentricity etc. Fig 4 legends: correct ‘PRE’ typo.

10. Line 332-334: Add graphical analysis of fluorescence intensity for Fig 5D (with sufficient N number). Replace with higher resolution (Fig 5B-D) and higher magnification images (for inner retina panel).

Scale bar info in legends seems incorrect for RPE panel.

11. Line 353: mention dose/duration for TNF-alpha treatment.

12. Fig 5B and 6A: Why the N number for Arpe19+estradiol were so low (almost half) as compared to control group? Estradiol group has a huge variation that questions the findings.

Where is the vehicle group for the in-vitro assay? Did the authors carry out validation of estradiol treatment in Arpe19 cells?

13. Add the transcriptional marker changes (RPE lysate) in different rat groups.

14. Line 356-357: It’s hard to say this. Not sure the increase/decrease seen in C5 and PGF are due to huge variation. Authors could resolve this by increasing the N number (for Estradiol group) and keeping N number same in both groups.

15. Same thing goes with IBA-1+ cell analysis (Fig 6D,E: in figures panel); huge variation.

16. Fig 6C: Macrophages are usually absent in the subretinal space unless in pathogenic state.

Considering Fig 6C: IBA-1+ cell IHC on RPE flat mounts, it is concerning that the 6mo old control rat showed large number of macrophages. This questions the health state of the 6mo control rat, which further questions the rest of the analysis.

Fig 6 legends: Correct 6C,D,E text information as per figure.

Add normal IHC images of IBA-1+ staining above AIVIA-based images.

**Do you want your identity to be public for this peer review?** For information about this choice, including consent withdrawal, please see our Privacy Policy

Reviewer #1: No

Reviewer #2: **Yes:** ZEESHAN AHMAD

Reviewer #3: **Yes:** Kiran Bora

---

## [Author Response · Author response to Decision Letter 1]

6 Nov 2025

In General

We have checked our manuscript for PlosOnes’ style requirements so that it now should fit into those.

2. Thank you for uploading your study's underlying data set. Unfortunately, the repository you have noted in your Data Availability statement does not qualify as an acceptable data repository according to PLOS's standards

We are using Zenodo as repository and have now loaded up all primary data sets. These data include supplemental files S1 to S3 with raw images showing original images of gel electrophoresis of Fig. 5, raw images used for AI based image analysis of Fig. 4 and 6, negative controls of the p16 and p21 staining of Fig. 3 and a table with the official antibody ID from Antibody Registry beta.

4. Thank you for stating the following financial disclosure: [Deutsche Forschungsgemeinschaft DFG Dr. Werner Jackstädt Stiftung Stiftung Charité in cooperation with Einstein Foundation Berlin].

We have included the funding acknowledgment statement in to the cover letter with the required detailed information.

5. PLOS ONE now requires that authors provide the original uncropped and unadjusted images underlying all blot or gel results reported in a submission’s figures or Supporting Information files. This policy and the journal’s other requirements for blot/gel reporting and figure preparation are described in detail at https://journals.plos.org/plosone/s/figures#loc-blot-and-gel-reporting-requirements and https://journals.plos.org/plosone/s/figures#loc-preparing-figures-from-image-files. When you submit your revised manuscript, please ensure that your figures adhere fully to these guidelines and provide the original underlying images for all blot or gel data reported in your submission. See the following link for instructions on providing the original image data: https://journals.plos.org/plosone/s/figures#loc-original-images-for-blots-and-gels. In your cover letter, please note whether your blot/gel image data are in Supporting Information or posted at a public data repository, provide the repository URL if relevant, and provide specific details as to which raw blot/gel images, if any, are not available. Email us at plosone@plos.org if you have any questions.

Along with our raw data, we have uploaded the file "S1_raw_images_Fig5" to zenodo, which shows the original images of the gel electrophoresis.

6. Please include captions for your Supporting Information files at the end of your manuscript, and update any in-text citations to match accordingly. Please see our Supporting Information guidelines for more information: In the published article, supporting information files are accessed only through a hyperlink attached to the captions. For this reason, you must list captions at the end of your manuscript file. You may include a caption within the supporting information file itself, as long as that caption is also provided in the manuscript file. Do not submit a separate caption file. Captions for supplemental data are added to the end of the manuscript.

Supporting material were now added, one for the request of gel demonstrations but also by suggestions of the reviewers. We uploaded these supplemental files into our repositorium. The file names and a short description of their content is added at the end of the manuscript. Furthermore, we made an additional comment in the “Data availability statement” to help readers to find them quickly.

There are some requests for the methods sections that we added .

Reviewer #1:

We have to thank the reviewer sharing our enthusiasm for our data. The critical remarks were very helpful and have led to a substantial improvement of our manuscript. The following responses/changes were given to his comments:

Main comment: The manuscript entitled “Evaluation of risk-promoting effects for age-related macular degeneration by estradiol” describes how biological or ovariectomy-induced estradiol deficiency promotes mechanisms that lead to age-related macular degeneration. Further, the authors documented, the estradiol-dependent promotion of cellular senescence is more pronounced than that of cellular inflammation. Importantly, the authors documented, the estradiol deficiency probably acts as a modulator that increases susceptibility to retinal degeneration rather than initiating a direct pathological cascade leading to age-related macular degeneration. Overall, the manuscript is well written with few errors. I have one major concern, although the authors have shown nice images of hematoxylin and eosin staining, exhibiting photoreceptor degeneration and a reduction in the ganglion cell nuclei count in ovariectomized rats. I suggest performing immunohistochemical analysis employing rod and cone photoreceptor antibodies to further confirm the photoreceptor degeneration.

This is basically a good suggestion made by the reviewer. Photoreceptor degeneration is defined by the loss of photoreceptor cells. The most precise way to measure this is counting nuclei in the rows of the ONL that includes both that of cones and rods. The counting of photoreceptors by their opsins is less precise as the high density of opsins make it difficult to separate individual cells. The suggestion to additionally evaluate immunohistochemical staining of rods and cones is even more difficult because rats have a rod-dominated photoreceptor layer. Due to the high density, rhodopsin staining to identify the photoreceptor outer segments would, based on experience, result in a single fluorescent line. For these reasons, it is the international standard to quantify photoreceptor degeneration by counting nuclei in the ONL; two examples Qi, H., Cole, J., Grambergs, R.C. et al. Sphingosine Kinase 2 Phosphorylation of FTY720 is Unnecessary for Prevention of Light-Induced Retinal Damage. Sci Rep 9, 7771 (2019). https://doi.org/10.1038/s41598-019-44047-z or Mirza, Myriam et al. “Progressive retinal degeneration and glial activation in the CLN6 (nclf) mouse model of neuronal ceroid lipofuscinosis: a beneficial effect of DHA and curcumin supplementation.” PloS one vol. 8,10 e75963. 4 Oct. 2013, doi:10.1371/journal.pone.0075963.

A few minor comments are below.

1. Introduction section, line 96. Incorporate a reference related to hormone therapy during menopause.

Thank you for the hint. We have added the clinical studies we wanted to cite. (line 94; (19-21), line 95; ROS (22, 23)).

2. Materials and methods section, line 143. Write the confluency of the cell culture as a percentage.

We added the information that for the experiments with semiconfluent cells, we used cultures in which 80% of the cells were confluent (line 143).

3. Materials and methods section, line 148. It should be “The primer sequences are provided in Table 1.”

Thank you for the improvement suggestion. We have implemented it exactly. (line 148 - 149)

4. Materials and methods section, line 150. Italicize the gene name throughout the manuscript, including the table.

We have corrected the style of gene names throughout the manuscript (including tables and figures) to gene names in italics and protein names in normal script. In the in vitro experiment with human cells, we used italicized capital letters. In the ex vivo experiment with rat samples, just the first letter capital and the names are also italic.

5. Figure legends 2, 3, and 5. Write the magnification of the images captured.

We have added magnification information to the legends of Figures 2 to 6. In addition, we have indicated in the Methods section that all images of paraffin sections were taken at 200- or 400-times magnification (line 224), and images of flatmount samples at 200-times magnification (line 239).

Reviewer #2:

We thank the reviewer for his very positive evaluation of or manuscript. He made had many important suggestions that we answered in the following ways:

This is a well-written and carefully executed study that investigates the contribution of estradiol deficiency to age-related macular degeneration (AMD) risk, with a focus on retinal degeneration, inflammation, and cellular senescence. The authors employed two rat models: naturally aged Lewis rats (22 months) to mimic biological estradiol decline and rats ovariectomized at 8 months, analyzed at 22 months, to model early menopause. Six-month-old rats served as controls. Methods included serum hormone analysis (ELISA, LC-MS/MS), qPCR for gene expression, immunohistochemistry for protein localization, AI-based image analysis for retinal cell density and inflammatory markers, and in vitro assays using ARPE-19 cells to assess ER function. The study provides compelling evidence that estradiol deficiency does not directly cause AMD but promotes retinal degeneration, inflammation, and senescence, thereby increasing AMD risk. Importantly, the data also point toward broader hormonal influences, with DHEA-S potentially contributing to photoreceptor degeneration in early menopause. While the manuscript is strong, several limitations should be addressed to strengthen its conclusions:

Comment 1: The manuscript provides convincing structural evidence of retinal alterations; however, the absence of functional vision assessment (e.g., ERG) limits translational impact. Electrophysiological characterization would determine whether hormonal deficiencies result in measurable visual dysfunction, thereby enhancing clinical relevance to AMD progression.

This is a very good suggestion, as also as we are able to record ERG. However, among our investigated animals are those of an age of 22 months. ERG recordings require anesthesia. As not only the retina is strongly affected, we expected severe side effects of ERG recordings such as reduced organ perfusion that will interfere with ERG signals apart from those caused by photoreceptor degeneration. Given the strong morphological affection of these retinas, we expect ERG recordings far away from normal base lines. This is also the reason for that the strengths of the ERG are to detect retinal dysfunction already before structural changes occur. Here, we want also emphasize that in the clinical practice assessments of retinal degeneration are rarely done by ERG. For the assessment of progression, the clinical findings refer exclusively to structural changes (for assessing functional changes, visual acuity testing with optotypes is the first choice) Thus, we found the structural analysis must suffice our conclusions.

Comment 2: The link between estradiol deficiency and p16/p21-driven senescence in ERβ-expressing retinal populations is compelling but requires deeper mechanistic insight. Single-cell RNA sequencing of retinal cell types (RGCs, INL cells, and RPE) from estradiol-deficient models could delineate cell type–specific senescence programs.

Thank you for the suggestion. We would be happy to implement this. However, single-cell RNA sequencing of retinal cell types (RGCs, INL cells, and RPE) from estradiol-deficient models is unfortunately not feasible within the period for a paper review: rats of age more than 20 months and final data after submission of the probes to the Charite Technology Platforms in at least three months. However, as the reviewer asks for, we find this extremely interesting and plan this our follow-up studies.

Comment 3: The study associates global hormone status (progesterone, corticosterone, DHEA-S, alongside estradiol) with retinal degeneration, inflammation, and senescence. However, the individual contributions of progesterone or corticosterone to the observed phenotype remain undefined. Clarifying their specific roles in retinal degeneration, inflammation, or senescence markers would strengthen the conclusions.

Thanks for this question. To our knowledge, this is may be the first paper that shows that menopause-dependent changes in the hormone system are not exclusively estrogen-dependent effects. The focus of our study is the role of estradiol but we want to put observations in relation to other possible effectors in menopause. We think researchers in this area should be aware of this. With the observed compensatory changes in other hormones, we wanted to emphasize the complexity of hormonal imbalance. Particularly because other papers conclude that any effects in the ovariectomy model are exclusively due to estradiol loss. Like the reviewer, we see as the next step the clarification of the role of other hormones affected by menopause. However, describing the precise effects of this imbalance of other sex hormones requires further studies of substantial coverage.

Comment 4: The use of 6-month-old rats as controls for both 22-month-old (aged) and ovariectomized 22-month-old rats may not adequately account for age-related changes independent of estradiol deficiency. An age-matched control group would better isolate the effects of estradiol loss from normal aging.

This is a question of the focus in our study. The 6 months old animals are controls to identify age-related changes in 22 months old normal rats. The comparison of 22 months old ovariectomized rats with 22 months old sham-operated controls reveal the differences by the disturbed hormone activity at age. The comparison of 6 months old sham-operated animals with 6 months old ovariectomized rats will reveal only the effects of hormone disturbance at young ages; may be a mistake in studies that use only young rats. Thus, our study focused on the impact of estradiol on aging processes rather than on estradiol's action per se. A link to AMD can only be evaluated in the changes in advanced animal age, together with estradiol deficiency.

Comment 5: The authors propose that photoreceptor degeneration and cellular senescence in ovariectomized rats may occur through ERβ-independent mechanisms, potentially mediated by DHEA-S reduction. However, this interpretation lacks direct experimental support. Additional validation—such as DHEA-S supplementation studies or downstream pathway analyses—would provide mechanistic clarity and strengthen claims about non-estradiol hormonal influences on AMD risk.

Yes, you are right, additional validation—such as DHEA-S supplementation studies or downstream pathway analyses—would provide mechanistic clarity and strengthen claims about non-estradiol hormonal influences on AMD risk. However, in our current study, the focus was on the impact of estradiol deficiency on the risk of AMD. The associated loss of DHEA-S should be mentioned here only for completeness, especially since it has been ignored in previous studies using the ovariectomy model. Further studies are urgently needed to unravel this complexity. We may also emphasize that such an interventional study would not be able in a normal period of paper revision but would also be a study of its own.

Reviewer #3:

We would like to thank also reviewer #3 for his detailed evaluation of the manuscript that revealed many items with the need for the correction. Our responses and changes in the manuscript are the following ones:

In this manuscript, authors have utilized two different rat models to study estradiol deficiency-related changes that could potentially be associated with AMD. S

---

## [Decision Letter · Decision Letter 1]

22 Dec 2025

Evaluation of risk promoting effects for age-related macular degeneration by estradiol

PONE-D-25-16542R1

Dear Dr. StrauB

We’re pleased to inform you that your manuscript has been judged scientifically suitable for publication and will be formally accepted for publication once it meets all outstanding technical requirements.

Kind regards,

Mohd Akbar Bhat, Ph.D.

Academic Editor

PLOS One

Additional Editor Comments (optional):

Reviewers' comments:

Reviewer's Responses to Questions

**Comments to the Author**

Reviewer #1: All comments have been addressed

Reviewer #2: All comments have been addressed

2. Is the manuscript technically sound, and do the data support the conclusions?

Reviewer #1: Yes

Reviewer #2: Yes

3. Has the statistical analysis been performed appropriately and rigorously?

Reviewer #1: Yes

Reviewer #2: Yes

4. Have the authors made all data underlying the findings in their manuscript fully available?

Reviewer #1: Yes

Reviewer #2: Yes

5. Is the manuscript presented in an intelligible fashion and written in standard English?

Reviewer #1: Yes

Reviewer #2: Yes

Reviewer #1: The authors have adequately addressed all my concerns including related to the immunohistochemical analysis.

Reviewer #2: (No Response)

**Do you want your identity to be public for this peer review?** For information about this choice, including consent withdrawal, please see our Privacy Policy

Reviewer #1: No

Reviewer #2: **Yes:** ZEESHAN AHMAD

---

## [Editor Report · Acceptance letter]

PONE-D-25-16542R1

PLOS One

Dear Dr. Strauß,

I'm pleased to inform you that your manuscript has been deemed suitable for publication in PLOS One. Congratulations! Your manuscript is now being handed over to our production team.

Kind regards,

on behalf of

Dr. Mohd Akbar Bhat

Academic Editor

PLOS One